# Stable maternal proteins underlie distinct transcriptome, translatome, and proteome reprogramming during mouse oocyte-to-embryo transition

Hongmei Zhang[1,2†], Shuyan Ji[1,2†], Ke Zhang[1,2†], Yuling Chen[3], Jia Ming[1,2], Feng Kong[1,2], Lijuan Wang[1,2], Shun Wang[4,5], Zhuoning Zou[1,2,6], Zhuqing Xiong[1,2], Kai Xu[1,2], Zili Lin[1,2], Bo Huang[7], Ling Liu[1,2], Qiang Fan[1,2], Suoqin Jin[4], Haiteng Deng[3] and Wei Xie[1,2*]

†Hongmei Zhang, Shuyan Ji and Ke Zhang contributed equally to this work.

*Correspondence:
xiewei121@tsinghua.edu.cn

[1] Center for Stem Cell Biology and Regenerative Medicine, MOE Key Laboratory of Bioinformatics, New Cornerstone Science Laboratory, School of Life Sciences, Tsinghua University, Beijing 100084, China
Full list of author information is available at the end of the article

## Abstract

**Background:** The oocyte-to-embryo transition (OET) converts terminally differentiated gametes into a totipotent embryo and is critically controlled by maternal mRNAs and proteins, while the genome is silent until zygotic genome activation. How the transcriptome, translatome, and proteome are coordinated during this critical developmental window remains poorly understood.

**Results:** Utilizing a highly sensitive and quantitative mass spectrometry approach, we obtain high-quality proteome data spanning seven mouse stages, from full-grown oocyte (FGO) to blastocyst, using 100 oocytes/embryos at each stage. Integrative analyses reveal distinct proteome reprogramming compared to that of the transcriptome or translatome. FGO to 8-cell proteomes are dominated by FGO-stockpiled proteins, while the transcriptome and translatome are more dynamic. FGO-originated proteins frequently persist to blastocyst while corresponding transcripts are already downregulated or decayed. Improved concordance between protein and translation or transcription is observed for genes starting translation upon meiotic resumption, as well as those transcribed and translated only in embryos. Concordance between protein and transcription/translation is also observed for proteins with short half-lives. We built a kinetic model that predicts protein dynamics by incorporating both initial protein abundance in FGOs and translation kinetics across developmental stages.

**Conclusions:** Through integrative analyses of datasets generated by ultrasensitive methods, our study reveals that the proteome shows distinct dynamics compared to the translatome and transcriptome during mouse OET. We propose that the remarkably stable oocyte-originated proteome may help save resources to accommodate the demanding needs of growing embryos. This study will advance our understanding of mammalian OET and the fundamental principles governing gene expression.

**Keywords:** Proteome, Translatome, Transcriptome, Oocyte-to-embryo transition, OET, Early embryo development

## Background

Remodeling the terminally differentiated gametes into a totipotent zygote enables the birth of life. This process is tightly controlled by maternally supplied RNAs and proteins accumulated during oogenesis [1, 2], as transcription is silenced at the end of the growing phase of oocyte until zygotic genome activation (ZGA) after fertilization [3–5]. Thus, meiotic resumption, fertilization, and early embryogenesis before zygotic genome activation are controlled by post-transcriptional regulation of maternal products. As a result, proteins are poorly predicted based on the transcriptomes as shown in *Xenopus* oocytes and early embryos [6]. Translation is expected to be more closely related to protein abundance as it reflects the protein-producing rate. Extensive transcription-independent translational regulation of maternal mRNAs occurs during the oocyte-to-embryo transition (OET) [7, 8]. For example, RNAs of key regulators can be pre-transcribed during oocyte growth but remain "dormant" (without being translated) and only resume translation upon meiotic resumption [7]. Inhibition of translation with cycloheximide (CHX) in mouse oocytes or early embryos led to severe developmental arrests [9–13]. Nevertheless, it remains unclear how well it predicts the total protein abundance, as the latter also depends on the existing protein level and the protein degradation rate. Currently, a full understanding of the fundamental relationship between transcriptome, translatome, and proteome that governs mammalian OET is still lacking.

Proteome investigations have been performed in mammalian gametes and early embryos [14–18]. In mice, previous studies have identified 3,000–6,550 proteins in mouse oocytes and early embryos with 600–8,000 oocytes or embryos at each stage, using a label-free approach [15], the tandem mass tags (TMT) method [16], or stable isotope labeling by amino acids (SILAC) approach [17]. On the other hand, due to highly limited experimental materials, the complete translatome data in mammalian preimplantation development were previously not available. Recently, we and others developed ultrasensitive translatome profiling methods and mapped the translatomes in mouse and human oocytes and early embryos [11, 12, 19, 20]. Here, using a label-free mass spectrometry strategy [21], liquid chromatography with tandem mass spectrometry (LC–MS/MS), we generated a high-quality proteome landscape in mouse oocytes and preimplantation embryos at the stages matching our transcription and translation datasets, using 100 oocytes or embryos per stage. By incorporating these three datasets, our study provides a unique opportunity to systematically investigate the relationship among the protein, translation, and transcription during mammalian OET. These data reveal distinct proteomes from transcriptomes and translatomes in oocytes and early embryos, shedding light on the multi-layered control of oocyte maturation and embryogenesis.

## Results

### LC–MS/MS efficiently quantifies the protein abundance

We sought to understand the relationship among the proteome, translatome, and transcriptome in mouse oocytes and early embryos. As previous proteome studies did not cover the full stages (Additional file 1: Fig. S1A) and it is not feasible to combine data from different methods, we first mapped the proteome landscapes using a highly sensitive label-free LC–MS/MS approach [21] at the stages matching those of the translatome and the transcriptome data we recently published [12]. To test its quantification capability, we first measured the protein abundance in FGOs with varying cell numbers (Fig. 1A). A total of 1,896, 3,072, 3,288, and 4,185 proteins

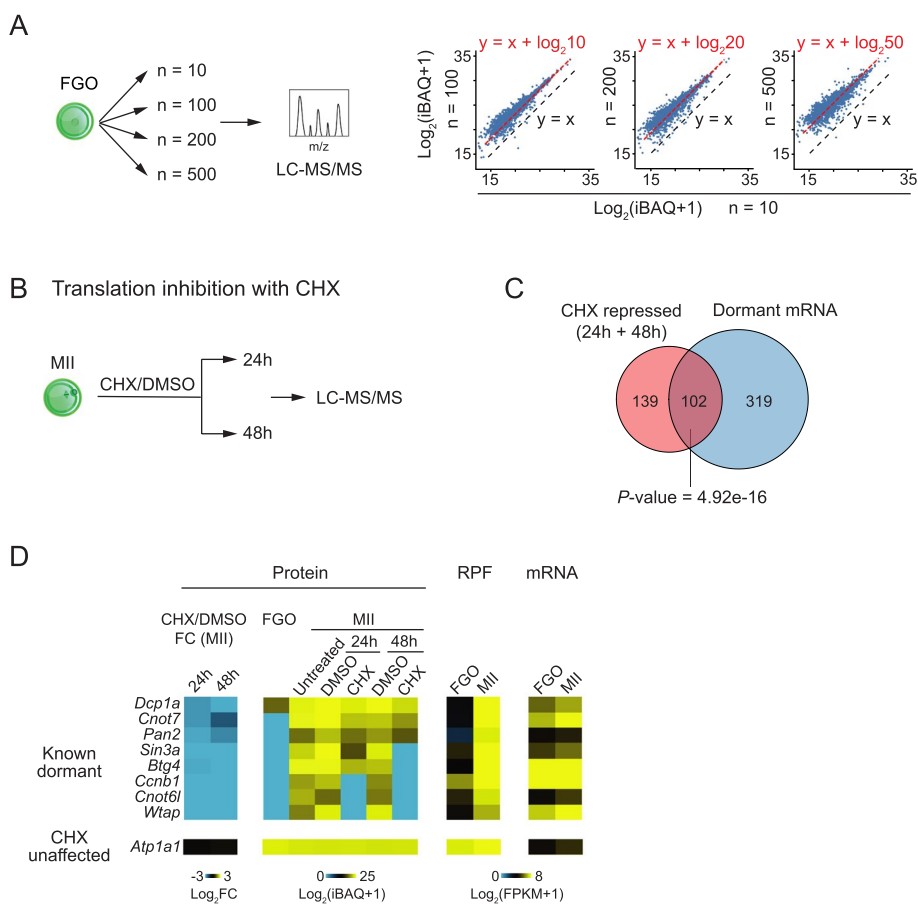

**Fig. 1** Low-input LC–MS/MS efficiently quantifies the protein abundance. **A** Validation of LC–MS/MS quantification capability. Schematic showing the experimental design of validation with various numbers of FGOs (left). Scatter plots comparing log₂ transformed protein intensities of 100, 200, and 500 FGOs against 10 FGOs (right). Lines representing the log₂ protein intensity with expected fold changes comparing with that of 10 FGOs are shown in red, and the reference line y = x is shown in black. iBAQ, intensity-based absolute quantification. **B** Schematic showing the experimental design of the MII oocytes treated with DMSO or CHX for 24 h or 48 h, followed by proteome profiling with LC–MS/MS. **C** Venn diagram showing the overlap of CHX repressed proteins and dormant mRNAs identified by Ribo-seq (based on ribosome protected fragments, RPF) [12] (RPF, MII oocyte/FGO > 2; mRNA, MII oocyte/FGO < 2). *P*-value calculated by Fisher's Exact test is also shown. **D** Heat maps showing protein fold changes (FC) and protein intensities upon CHX treatment in MII oocytes for known dormant genes. RPF and mRNA levels are also shown. A CHX unaffected gene is shown as a control

were identified from 10, 100, 200, and 500 FGOs, respectively (Additional file 1: Fig. S1B). Proteins consistently detected in all FGO samples ($n = 1{,}813$) were then used to evaluate the quantification capability of LC–MS/MS. Compared to the protein abundance from 10 FGOs, that from 100, 200, and 500 FGOs increased correspondingly by roughly 10-, 20-, and 50-fold, respectively (Fig. 1A, right), demonstrating the superior quantification ability of our results. For further validation, we applied the protein synthesis inhibitor cycloheximide (CHX) to oocytes (Fig. 1B), where a number of dormant RNAs are known to be newly translated during FGO to MII oocyte maturation. CHX also did not affect the development of MII oocytes which are naturally arrested prior to fertilization. We did not choose FGOs as their maturation is blocked upon the CHX treatment [22, 23], and the differential translation would be confounded by the maturation defects. We observed 241 proteins down-regulated (24 h and 48 h treatments were combined  due to their similarity) compared to the DMSO control (Additional file 1: Fig. S1C, top). Consistently, these proteins were poorly detected in FGOs but were highly abundant in untreated MII oocytes (Additional file 1: Fig. S1C, top), suggesting they were newly synthesized. A comparison with our Ribo-seq data revealed 102 (42.3%) of the 241 CHX-repressed proteins were translated from dormant mRNAs [12] which were lowly translated in FGOs but were highly translated in MII oocytes (at least two-fold upregulated) (Fig. 1C and Additional file 1: Fig. S1C, top), such as *Cnot7*, *Pan2*, *Sin3a*, and *Btg4* (Fig. 1D). CHX-repressed proteins were significantly enriched for genes involved in cell cycle (e. g., *Aurka*, *Cenpe*, and *Rad51*), RNA stability (e.g., *Cnot6l*, *Cnot7*, and *Pan2*), chromatin organization (e.g., *Ezh2*, *Sin3a*, and *Tet3*), and histone modification (e.g., *Hdac1/2*, and *Kdm1a/b*) pathways (Additional file 1: Fig. S1D). Many of these genes are key regulators in meiosis resumption, maternal mRNA clearance, and epigenetic reprogramming during oocyte maturation. By contrast, CHX-unaffected proteins tended to be highly translated in FGOs, but are lowly translated in MII oocytes (hence insensitive to CHX) (Additional file 1: Fig. S1C, bottom). These proteins showed enrichment in housekeeping pathways, such as the amide metabolic process and intracellular protein transport (Additional file 1: Fig. S1D). Furthermore, the proteome measurements aligned well with our previous Western blot analyses [12] (Additional file 1: Fig. S1E). Overall, these data demonstrate that our proteome data are highly quantitative and sensitive for low-input samples.

### Dynamic proteome distinct from the translatome and the transcriptome in oocytes and embryos

Next, we performed systematic profiling of protein abundance in FGO, MII oocyte, 1-cell (1C), 2-cell (2C), 4-cell (4C), 8-cell (8C), and blastocyst (BL) embryos, using 100 oocytes or embryos for each developmental stage (Fig. 2A). We collected the oocytes and embryos developed in vivo up to the 8C stage, and obtained blastocysts through in vitro culturing starting from the 8C embryos for the convenience of sample collection. It is worth noting that superovulation and in vitro culturing may influence the qualities of mouse embryos and their proteomes [24]. Early embryos were obtained from crosses of C57BL/6J female and PWK/PhJ male mice to match the genetic background of our translatome and transcriptome data [12]. Protein abundance

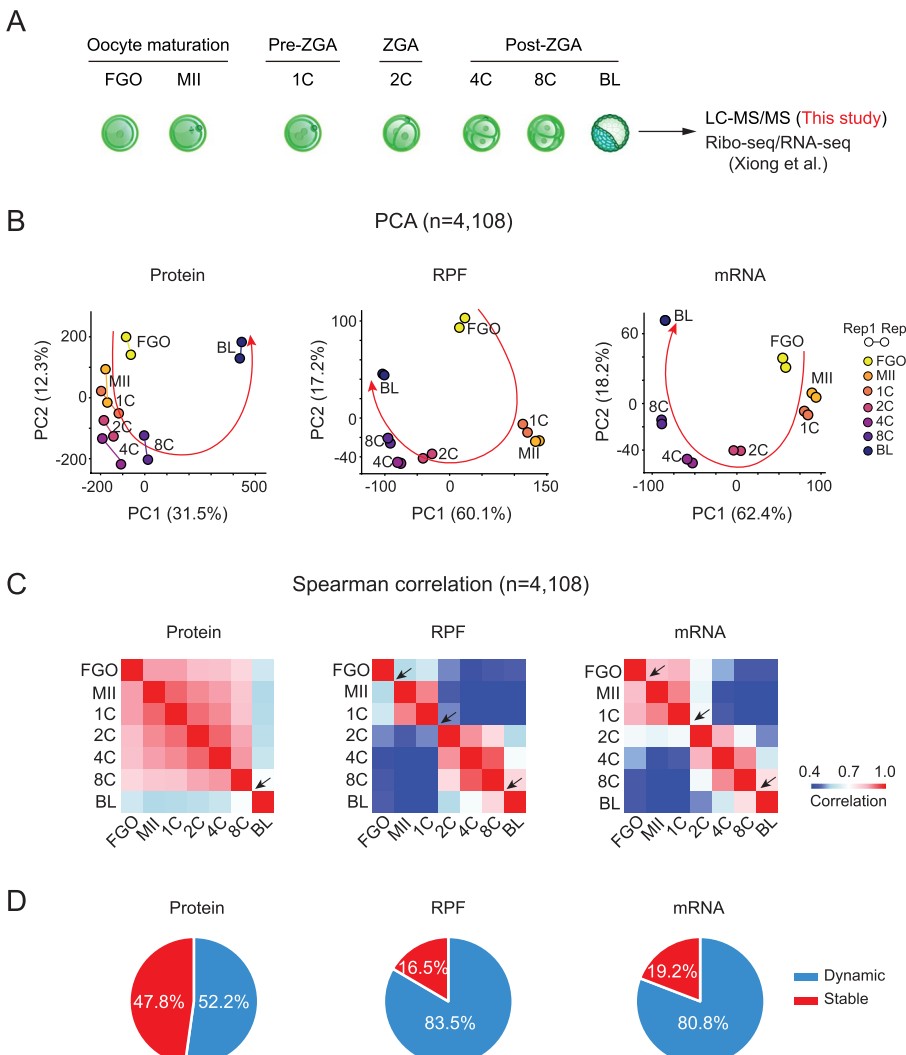

**Fig. 2** Distinct global proteome, translatome, and transcriptome dynamics during mouse oocyte maturation and early development. **A** Schematic of mouse oocytes and preimplantation embryos used for LC–MS/MS, Ribo-seq, and RNA-seq. **B** Principal component analysis (PCA) of genes ($n = 4,108$) based on their protein, translation (RPF), and transcription (mRNA) levels. The sample points are colored by developmental stage and the replicates are connected with solid lines. The developmental trajectories of samples are shown in red lines. **C** Heat maps showing the Spearman correlation coefficients between pairwise developmental stages for protein, RPF, and mRNA. Arrows indicate the stages that show evident transitions for the proteome, translatome, or transcriptome. **D** Pie charts showing the proportions of stably and dynamically regulated (coefficient of variation > 0.2 of log2 transformed intensity across stages) proteins, RPFs, and mRNAs for detected genes ($n = 4,108$) in oocytes and early embryos

measurements were highly reproducible between replicates ($R = 0.79$–$0.90$) (Additional file 1: Fig. S2A), and a total of 5,933 proteins were detected (Additional file 2: Table S1). Compared to previous work [15–17], our study detected comparable proteins with the fewest cells (Additional file 1: Fig. S2B, C) and covered the most comprehensive stages during OET (Additional file 1: Fig. S1A). After correcting batch effects (Methods), 4,108 proteins that were consistently detected across batches and also detected in our Ribo-seq (based on ribosome protected fragments, RPF) and RNA-seq (mRNA) datasets [12, 25] were subsequently used for the downstream

analysis (Additional file 3: Table S2). We then compared the temporal dynamics of the proteome, translatome, and transcriptome across developmental stages. Principal component analysis (PCA) and pairwise Spearman correlation analysis revealed the translatome and the transcriptome showed multiple transitions along the development trajectory (Fig. 2B, C). Major transitions were observed at the FGO-to-MII, 1C-to-2C, and 8C-to-blastocyst stages (Fig. 2B, C, arrows), aligning with three main events at these stages: meiotic resumption, zygotic genomic activation, and the first lineage specification, respectively. By contrast, the proteome was much less dynamic from FGO to 8C, before major transitions occurred during the 8C-to-blastocyst (Fig. 2B, C, arrows), consistent with the previous findings [16, 17]. Further coefficient of variation (CV) analysis (Methods) revealed the translation of 83.5% of genes and the transcription of 80.8% of genes were dynamically regulated throughout the course of oocyte maturation and early development (Fig. 2D). This difference is not due to bias introduced by previous gene filtering, as similar fractions of variable genes were identified for the translatome and the transcriptome using all expressed genes (without requiring that they had to be detected by mass spectrometry, Additional file 1: Fig. S3). By contrast, only 52.2% of genes showed dynamic proteins in oocytes and early embryos.

We next investigated the temporal dynamics of individual proteins across developmental stages. The K-means clustering analysis classified the detected proteins into six distinct groups (Fig. 3A, Additional file 3: Table S2). 1) The largest cluster included "Constitutive" proteins which maintained high abundance across all developmental stages despite the dynamic changes at the RPF and mRNA levels (Fig. 3A, B, "Constitutive"). These proteins could be further classified based on their RPF and mRNA dynamics. In the 1st subgroup, the RPF and mRNA abundance were maintained at high levels across all stages similar to protein dynamics (Fig. 3A, "protein constitutive-RPF constitutive"), including genes that are involved in RNA splicing (e.g., *Ddx5* and *Hnrnp*) and cell cycle (e.g., *Pcna* and *Mcm2*). The genes in 2nd subgroup were transcribed and translated in FGOs but were downregulated from the MII oocytes before they were reactivated after ZGA (Fig. 3A, "protein constitutive-RPF OET downregulated"). These genes were mainly involved in translation initiation (e.g., *Rps* and *Rpl*) and oxidative phosphorylation (e.g., *Ndufs*). The 3rd subgroup genes were lowly transcribed and translated in oocytes and early embryos (Fig. 3A, "protein constitutive-RPF low"), which are involved in small molecule catabolism (e.g., *Gapdh*) and Golgi vesicle transport (e.g., *Golga4*). These proteins were likely accumulated during oocyte growth. The 4th subgroup included genes translated in oocytes and pre-ZGA embryos (2C-stage) but strongly downregulated afterward (Fig. 3A, protein constitutive-RPF maternal). The Rho GTPases signaling and nuclear envelope assembly genes were enriched in this group.

2) "Maternal" proteins were highly expressed from FGO to MII oocytes and gradually degraded after fertilization with concordant transcription and translation dynamics (Fig. 3A, B, "Maternal"). These genes were primarily involved in oocyte development and fertilization (e.g., *Cpeb1*, *Gnrh1*, *Izumo1r*, and *Tdrd5*).

3) "OET downregulated" proteins were highly expressed in FGOs, but downregulated either from the MII oocytes or from the 2C embryos, before they reappeared in the 8C embryos or blastocysts (Fig. 3A, B, "OET downregulated"). Proteins in this class included

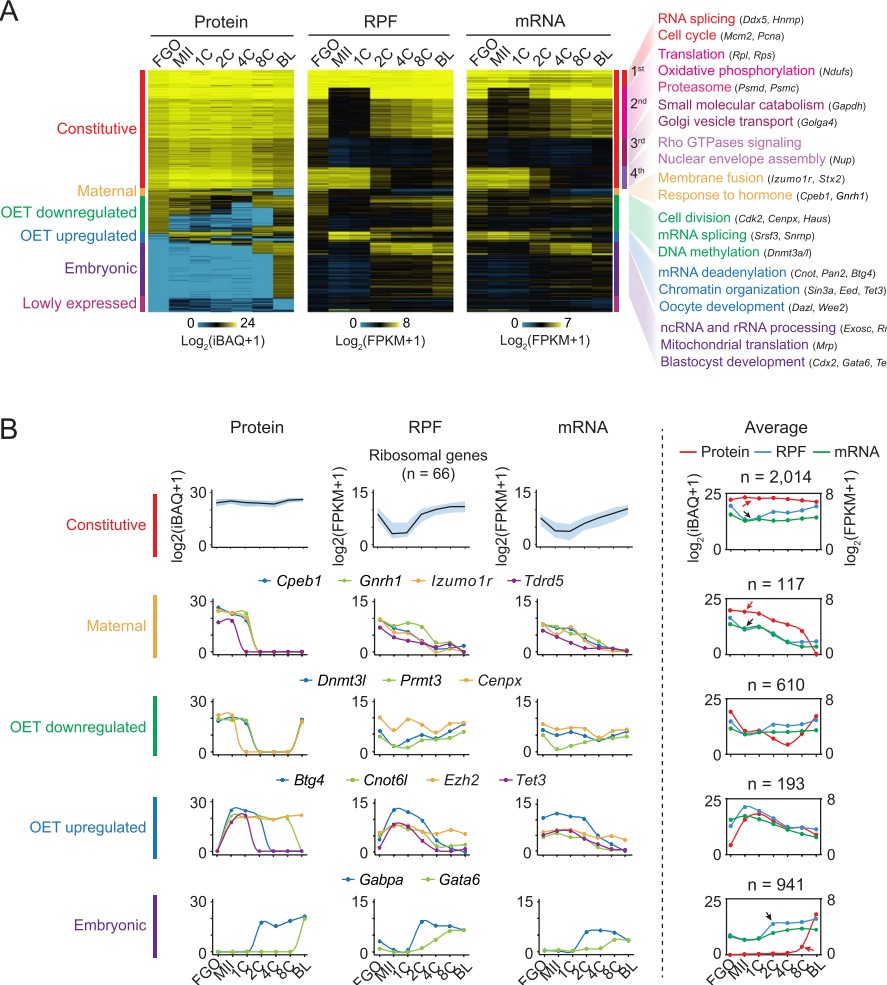

**Fig. 3** Global proteome dynamics in mouse oocytes and pre-implantation embryos. **A** Heat maps showing the K-means clustering results based on protein dynamics in oocytes and early embryos ($n = 4,108$), with the corresponding RPF and mRNA levels mapped. The enriched GO terms and example genes are also listed. Constitutive proteins are further classified into four subgroups based on the dynamics of RPFs. 1st subgroup, protein constitutive-RPF constitutive; 2nd subgroups, protein constitutive-RPF OET downregulated; 3rd subgroup, protein constitutive-RPF low; 4th subgroup, protein constitutive-RPF maternal. **B** Line plots showing the protein, RPF, and mRNA dynamics across stages for genes from different clusters in (A). Left, the representative genes or gene families; right, the average signals for each cluster. Arrows indicate discordant dynamics among proteome (red), translatome (black), and transcriptome (black) datasets

DNMT3A and DNMT3L, consistent with the global DNA demethylation after fertilization and global re-methylation starting from the blastocyst [26]. They also included factors involved in mRNA splicing (e.g., snRNPs). Notably, inhibiting splicing could convert mESCs to totipotent blastomere-like cells (TBLC) [27] or 2-cell-like cells [28].

4) "OET upregulated" proteins were present at low levels in FGO but were up-regulated upon meiotic resumption. The translation dynamics were consistent with protein changes exhibiting features of "dormant RNAs" (Fig. 3A, B, "OET upregulated"). These genes include regulators of mRNA deadenylation (e.g., *Btg4,* and *Cnot6l*) and chromatin organization (e.g., *Sin3a, Ezh2, Eed,* and *Tet3*) (Fig. 3B, "OET upregulated"). The upregulation of TET3 upon meiotic resumption is consistent with its role in the upcoming

global DNA methylome demethylation [29] and also explains why it does not demethylate the oocyte genome. The upregulation of EED and EZH2, components for Polycomb repressive complex 2 (PRC2), is interesting as oocyte-inherited H3K27me3 enables the allele-specific expression of key imprinted genes (e.g., *Gab1*, *Sfmbt2*, and *Slc38a4*) and X chromosome genes (*Xist*) in early embryos [30–32].

5) "Embryonic" proteins are translated after ZGA, encoded by genes that are involved in ribosome biogenesis (e.g., *Exosc* and *Rrp*), histone modification (e.g., *Kdm3a/b*), and blastocyst development (e.g., *Gata6*, *Gabpa*, and *Cdx2*) (Fig. 3A, B, "Embryonic"). Notably, genes with stronger translation and transcription levels tended to show detectable proteins at the earlier stages (2-8C), while genes with weaker translation and transcription did not exhibit detectable proteins until the blastocyst stage, likely due to the limited sensitivity of LC–MS/MS.

6) The last group included "Lowly expressed" proteins with low levels of RPFs and mRNAs, which were excluded from subsequent analyses.

Therefore, these results indicate that the proteome, translatome, and transcriptome undergo distinct reprogramming during oocyte maturation and embryogenesis. The proteome is less dynamic than the translatome and the transcriptome, with a large portion of proteins remaining stable even when translation and transcription exhibit substantial changes.

### FGO-8C proteomes are dominated by FGO-originated proteins

To better understand why the proteomes are distinct from the translatomes and the transcriptomes during OET, we first performed a systematic comparison at each stage. The correlations between translatome and transcriptome were high throughout development ($R = 0.78$–$0.90$) (Fig. 4A). By contrast, the proteome correlated poorly with the translatome ($R = 0.27$–$0.44$) and the transcriptome ($R = 0.26$–$0.41$), except in FGOs ($R = 0.70$ and $0.75$, respectively) and blastocysts ($R = 0.50$ and $0.68$, respectively) (Fig. 4A). The discrepancy between the proteome and the translatome or the transcriptome from MII oocyte to 8C was evidenced by the downregulation of translation and transcription for "constitutive proteins" (Fig. 3B, "Constitutive", red and black arrows), the delayed degradation of maternal proteins (Fig. 3B, "Maternal", red and black arrows) and the slow accumulation of embryonic proteins (Fig. 3B, "Embryonic", red and black arrows) relative to transcription and translation.

Strikingly, pairwise correlation analysis between developmental stages showed that the proteomes from FGOs to 8C embryos were all well correlated with the translatome of FGOs, though the correlation gradually decreased (Fig. 4B, left black box). These correlations (e.g., 4C proteins vs FGO RPFs) were often much higher than those of the proteome-translatome at the same stages (e.g., 4C proteins vs 4C RPFs). A similar observation was made for the proteome-transcriptome analysis (Additional file 1: Fig. S4A, left black box). These results suggest the proteomes in oocytes and early embryos are dominated by the "FGO proteome", likely due to the long-lasting proteins stockpiled in FGOs. In fact, among 103 strictly defined FGO-originated proteins with no or low translation after fertilization (RPF FPKM < 5 at the 1C stage and afterward), 58.3% of them were still detectable in blastocyst, such as CCDC136, SIRT5 and MTHFD1L (Fig. 4C), suggesting that the degradation of maternal proteins, in general, is slow in early embryos.

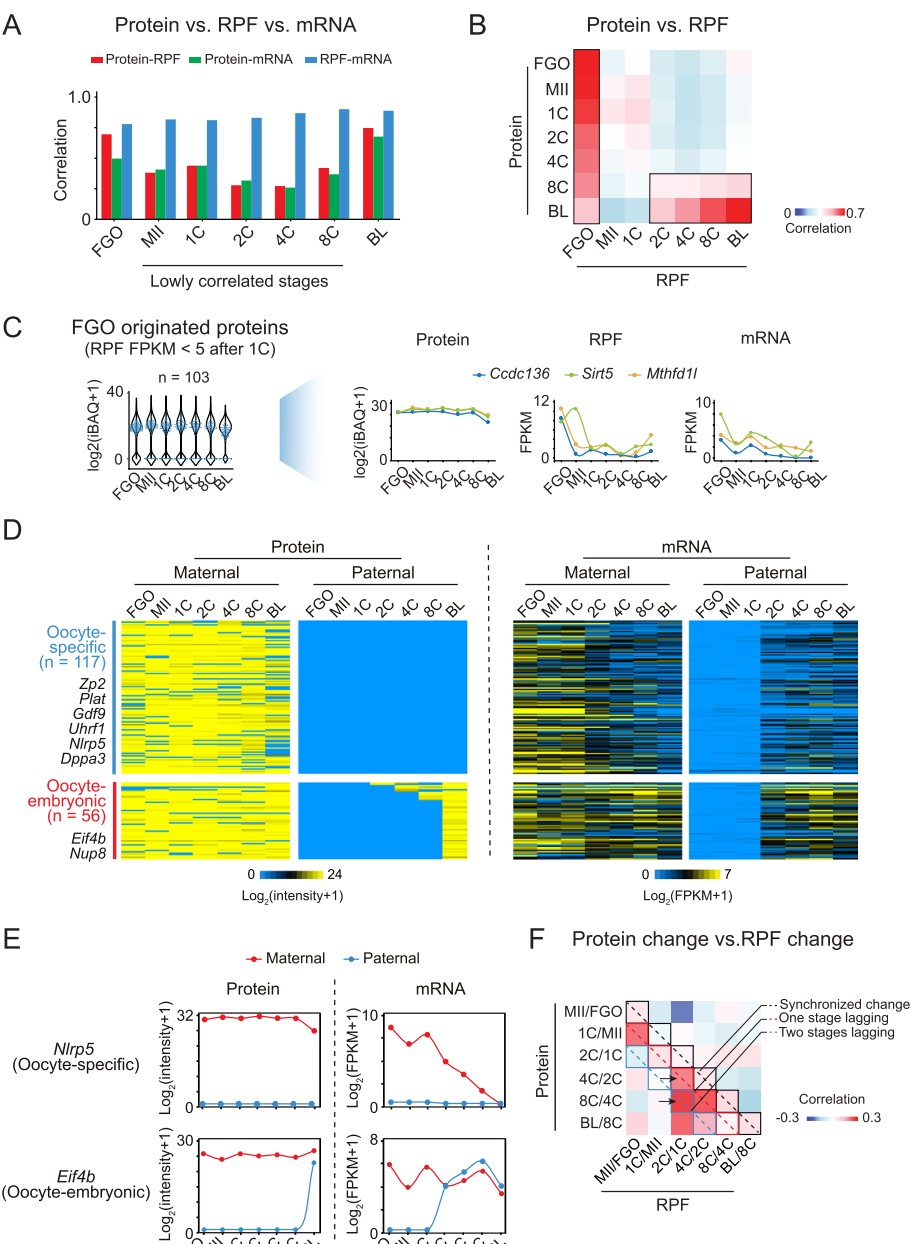

**Fig. 4** Correlation among protein, RPF, and mRNA during oocyte maturation and early embryo development. **A** Bar plots displaying the Spearman correlation coefficients between protein-RPF, protein-mRNA, and RPF-mRNA at each developmental stage. **B** Heat map showing the Spearman correlation coefficients between pairwise protein and RPF among different developmental stages. The black boxes indicate the highly-correlated stages. **C** Left, violin plot showing the protein levels of all proteins ($n = 4,108$). Blue dots are proteins that originated from FGOs but were not translated or lowly translated in early embryos (FGO originated, RPF FPKM < 5 at the 1C stage and afterward). Right, line plots show the protein, RPF, and mRNA dynamics of representative genes. **D** Heat maps showing the parent-of-origin dynamics of proteins in oocytes and early embryos, with the corresponding parent-of-origin mRNA levels mapped. The example genes are also listed. maternal, C57BL/6J; paternal, PWK/PhJ; $n$, gene number. **E** Line plots showing the parent-of-origin of proteins and mRNAs across developmental stages for example genes. **F** Heat map showing the Spearman correlation coefficients between protein changes and RPF changes among different developmental stages. The dashed lines represent synchronized (black), one-stage lagging (red), and two-stage lagging (blue) changes. Arrows indicate the example transition stages that are highly correlated between protein changes and RPF changes

As a separate validation to the analyses above, the highly variable genome sequences between C57BL/6J and PWK/PhJ strains provide an opportunity to investigate parent-of-origin dynamics of oocyte-originated and embryonic proteins. A total of 484 proteins with non-synonymous protein variations caused by SNPs were identified by LC–MS/MS, among which 173 variant proteins showed differential expression between the two alleles at one or more developmental stages in early embryos (Fig. 4D, Additional file 4: Table S3). Among them, 117 proteins were likely maternal proteins that originated from the oocyte and persisted in early embryos with no paternal-originated protein detected (Fig. 4D and E, "oocyte-specific"), including known maternal factors such as ZP2, GDF9, DPPA3, NLRP5 and PLAT. Consistently, their paternal RNAs were generally low in early embryos, suggesting the absence of zygotic transcription. Oocyte-originated transcript levels also reduced abruptly after ZGA. In addition, 56 proteins displayed oocyte expression and persisted until the embryonic stage, with paternally-derived proteins also detected at certain stages after ZGA, mostly at the blastocyst stage (Fig. 4D and E, "oocyte-embryonic"). In agreement with the protein dynamics, transcripts of maternal origin were present throughout oocytes and early embryos, and transcripts of paternal origin emerge after ZGA. We also identified 21 and 4 proteins that were only detected in embryos and showed exclusively maternal and paternal origin, respectively. However, the majority of them showed biallelic mRNA expression in embryos, raising the possibility that they could be false positives due to detection dropout in LC–MS/MS. Overall, these results indicate the FGO-originated proteins can persist until blastocyst, even in cases where their mRNAs have been degraded.

Interestingly, the translatome and transcriptome after ZGA also poorly correlated with the proteome at the same stage, but strongly correlated with the proteomes of the blastocyst and, to a lesser extent, 8C embryos (Fig. 4B and Additional file 1: Fig. S4A, right black box). This finding is consistent with the above observation that many genes were transcribed from the 2C stage, yet their proteins were not detected until the 8C or blastocyst stage (Fig. 3A and 4D). Indeed, when analyzing the changes between consecutive stages, protein changes after ZGA often correlated with RPF/mRNA changes at the one or two preceding stages (Fig. 4F and Additional file 1: Fig. S4B, red and blue dashed line). For example, the protein changes between the 2C-to-4C and the 4C-to-8C transition better correlated with translation and transcription changes from the preceding 1C-to-2C transition than those changes at the same stage (Fig. 4F and Additional file 1: Fig. S4B, black arrows). Therefore, the detectable protein changes lag behind the translation and transcription changes, which likely reflects both the latencies associated with protein synthesis and maturation [33] and the limited sensitivity of LC–MS/MS.

### Protein changes were partially contributed by transcription and translation changes

Despite the relatively stable proteome, we then focused on proteins that did show changes along oocyte development and embryogenesis and asked to what extent could transcription and translation changes explain their dynamics. To do so, we identified the up- and down-regulated proteins, RPFs, and mRNAs between each of the two consecutive stages. Among the 4,108 proteins, 1,983 (48.3%) proteins showed differential abundance in at least one stage transition ($|$fold change$| \geq 2$ and *P-value* $< 0.05$)

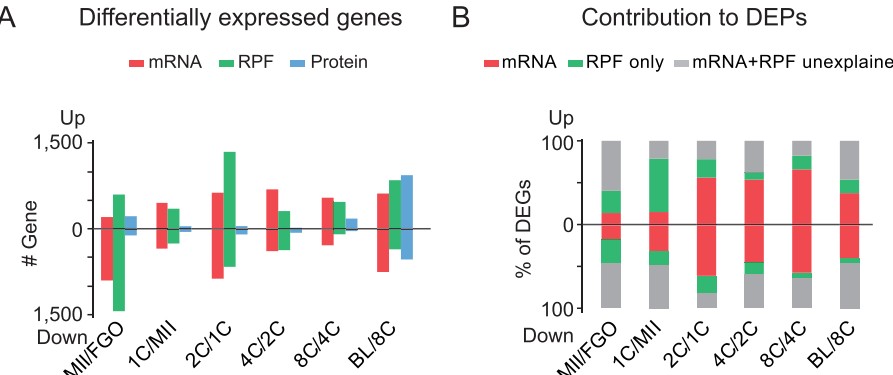

**Fig. 5** Analyses of differentially expressed genes during oocyte maturation and early embryo development. **A** Bar plots showing the numbers of differentially expressed mRNAs, RPFs, and proteins between consecutive developmental stages (|fold-change| ≥ 2 and *P*-value < 0.05 with Student's t-test). **B** Bar plots showing the percentages of differentially expressed proteins (DEPs) that could be explained by mRNA changes, additional RPF changes, or neither at each pair of consecutive developmental stages. mRNA, |mRNA change| > two-fold; RPF only, |RFP change| > two-fold and |mRNA change| < two-fold; mRNA + RPF unexplained, |mRNA change| < two-fold and |RPF change| < two-fold

(Fig. 5A, Additional file 5: Table S4). Only small numbers of differentially expressed proteins (DEPs) ($n = 85$–$334$) were identified at each consecutive transition before the 8C stage. The 8C-to-blastocyst transition displayed the largest number of DEPs (938 up-regulated and 542 down-regulated) (Fig. 5A), accordant with the reported major transition period [16, 17]. DEPs before ZGA were mainly involved in RNA splicing, cell cycle, and protein localization (Additional file 1: Fig. S5A). DEPs after ZGA were associated with small molecule catabolism, DNA metabolism, ribosome biogenesis, and cell pluripotency (Additional file 1: Fig. S5A). These terms are consistent with the metabolic switch in the early embryos [34] and the establishment of pluripotency in the blastocyst [35]. Meanwhile, much larger numbers of genes ($n = 579$–$2{,}044$) showed differential RPFs and mRNAs at each consecutive stage (Fig. 5A) and only small percentages of them (6.1%-31.8%) overlapped with DEPs (Additional file 1: Fig. S5B, C), suggesting the mRNA and RPF changes do not necessarily lead to significant changes in proteins. We next sought to estimate the contribution of translation and transcription to protein dynamics. As protein alteration was delayed relative to changes in translation and transcription (Fig. 4F and Additional file 1: Fig. S4B), the DEGs of mRNA and RPF from the current and the preceding two stages were all considered. If the protein changes were accompanied by corresponding mRNA changes, we considered that such protein changes are likely explained by mRNA changes (mRNA contributed). In most cases (93%), mRNA changes are also associated with RPF changes. For the rest protein changes, if they were not accompanied by apparent mRNA changes but were associated with RPF changes, we considered them as "RPF-only contributed", in which protein dynamics are potentially explained by mRNA-independent translation regulation. The results showed changes of 43～63% DEPs could be explained by changes in either mRNA or RPF (|mRNA change| > 2 or |RPF change| > 2) before ZGA (Fig. 5B). The up-regulation of proteins before ZGA

was mainly achieved by increasing translation efficiency (RPF-only contributed, 27–63%). After ZGA, the changes in mRNA expression could explain the variance of 39~59% of proteins, while the additional translational control accounted for a relatively small extra portion (11~21%) of protein variance (Fig. 5B). The rest changes in protein abundance (20~57%) were independent of the counterparts in mRNA or RPF (|mRNA change| < 2 and |RPF change| < 2, Fig. 5B), indicating possible post-translational regulation. Taken together, these results show that translational and post-translational controls likely contribute substantially to protein dynamics prior to ZGA, while transcriptional regulation contributes most to protein changes after ZGA.

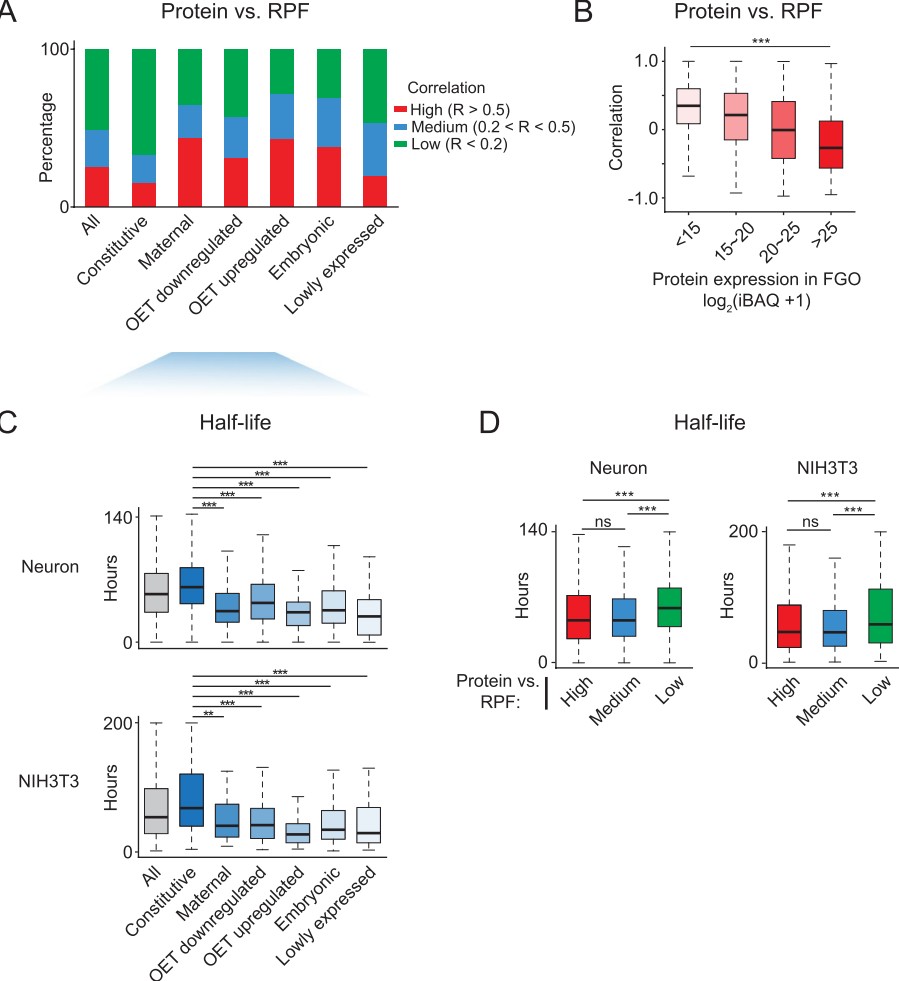

**Fig. 6** Protein-RPF concordance is linked to the initial protein abundance and the protein half-life. **A** Bar plots showing the percentages of proteins that showed high ($R > 0.5$), medium ($R > 0.2$ and $R < 0.5$), or low ($R < 0.2$) correlation (Spearman) with RPF for "All" proteins ($n = 4,108$) and different protein groups in Fig. 3A. **B** Box plots showing the protein-RPF correlation based on the protein abundance in FGOs. **C** Box plots showing half-lives determined in mouse embryo neuron [37] or embryonic fibroblast cell line NIH3T3 [36] for different protein groups. **D** Box plots showing protein half-lives for different protein-RPF correlation groups. The significance for all plots was calculated by Wilcoxon rank-sum test (two-tailed). ***, *P*-value < 0.001; **, *P*-value < 0.01; ns, non-significant

**Protein concordance with translation and transcription correlates with its oocyte stock level and half-life**

We were curious about what protein features would predict better correlations between protein and translation or transcription during development. Therefore, we calculated the protein-mRNA and protein-RPF correlation for individual genes across developmental stages (gene-wise correlation, Fig. 6A and Additional file 1: Fig. S6A, B). The correlation differed widely among genes, with median values of 0.18 and 0.19 for protein-RPF and protein-mRNA, respectively (Additional file 1: Fig. S6A, B). Correlations based on protein-mRNA and protein-RPF were largely consistent ($R = 0.76$) (Additional file 1: Fig. S6C, left), with a small group of them (29.2%) showing discordance (Additional file 1: Fig. S6C, pink and blue shades). For example, OET upregulated and embryonic proteins tended to show better protein-RPF correlation than protein-mRNA (Additional file 1: Fig. S6D, E, left). This was consistent with OET-upregulated genes being upregulated for translation without changing mRNAs upon meiotic resumption. By contrast, constitutively expressed proteins were better correlated with mRNA than RPF as their proteins remained stable despite downregulation of translation upon meiotic resumption (Additional file 1: Fig. S6D, E, right). Note that such a higher correlation between mRNA-protein compared to RPF-protein does not contradict a closer relationship between translation and protein, but rather caused by the stable protein and mRNA levels due to different reasons (a large amount of stable FGO protein and the lack of transcription during OET, respectively). Overall, about 25% of proteins showed high correlations with translation ($R > 0.5$, high), 24% of proteins showed modest correlations ($R = 0.2$–$0.5$, medium), and the rest 51% of proteins were lowly correlated ($R < 0.2$, low) (Fig. 6A, "All"). A similar trend was observed for the correlation of protein-mRNA (Additional file 1: Fig. S7A, "All"). As expected, "constitutive proteins" displayed the lowest correlation. This can be attributed to the fact that the RPFs and mRNAs for a significant proportion of them were downregulated from MII oocytes to early embryos (Fig. 3A, RPF for constitutive proteins), while new proteins were continuously produced, albeit at a growingly slower speed. As a result, the correlation between RPF/mRNA and protein even became negative (Additional file 1: Fig. S7B-C, 2nd and 3rd subgroups). By contrast, proteins of maternal, OET up- and down-regulated, and embryonic groups showed better correlations with translation and transcription (Fig. 6A and Additional file 1: Fig. S7A), consistent with the notion that developmental genes are under rapid control while housekeeping genes are subjected to protein "buffering" to transcript fluctuations [36] (see Fig. 3B for examples).

Notably, constitutive proteins are both abundant in FGOs and are stably present throughout development (Fig. 3A), partially explaining the low protein-RPF correlation. Indeed, genes showing the highest protein levels in FGOs had the lowest protein-RPF correlations (Fig. 6B and Additional file 1: Fig. S7D). Two possible reasons for the remarkable stability of "constitutive proteins" could be envisioned. 1) These proteins were accumulated in large quantities during oocyte growth. As a result, de novo protein synthesis after meiotic resumption only contributed to a small portion of the total protein and hence did not substantially affect its total levels. 2) These proteins may be resistant to degradation. Currently, no protein half-life datasets are available in mouse oocytes and early embryos. Given protein half-lives are highly conserved among cell

types [37–39], we utilized two protein half-life datasets from mouse embryonic neurons [37] and mouse embryonic fibroblast cell line NIH3T3 [36]. Indeed, constitutive proteins showed the longest half-lives among all classes (Fig. 6C). Globally, proteins showing high or medium correlation with RPF had shorter half-lives, while those with low correlation had longer half-lives (Fig. 6D and Additional file 1: Fig. S7E). It was reported that proteins with intrinsically disordered regions (IDRs) had shorter half-lives than those without IDRs [40]. Indeed, we found proteins with high protein-RPF or protein-mRNA correlations are more likely to contain IDRs than those with low correlations (Additional file 1: Fig. S7F). Taken together, the high abundance and long half-life of protein likely contribute to the discordance between protein and translation or transcription during mouse oocyte maturation and early embryo development.

**Protein dynamics could be predicted by translation and the initial stock protein abundance**
Given the discordance of protein and RNA/RPF, we next asked if we could predict protein dynamics given the translation profile and the initial protein abundance in FGO. Based on a mass-action kinetics model for the protein dynamics prediction [6, 41], the expected protein abundance at a specific time was determined by the initial protein abundance in FGO ($P_0$), the newly synthesized protein level, and also the degraded protein level over time (Fig. 7A, "$P_0+$ RPF model"). The newly synthesized protein was calculated by RPF level times a protein-specific constant α. To simplify the model, we assumed the degradation rate $k_d$ for each protein to be invariant across development. For each protein, the parameters $α$ and $k_d$ could be inferred by minimizing the average difference between the observed and predicted proteins across all time points (Methods). The results showed the $P_0+$ RPF kinetic model could well predict the protein dynamics throughout development (Fig. 7B), as exemplified by genes *Rps21* and *Btg4* (Fig. 7C). Notably, the $P_0+$ RPF model, but not the RPF-only model which did not consider the initial protein abundance, could well predict the dynamics for constitutive proteins, especially at stages when the translation was repressed (Fig. 7B, red boxes), supporting the buffering effects of existing proteins against translation and transcription perturbation. The median correlation between predicted and measured protein abundance was 0.62 for the $P_0+$ RPF model, a striking improvement over the RPF only model ($R = 0.23$) or the simple protein-RPF correlation (median $R = 0.18$) (Fig. 7D). These results suggest that our simple kinetic model with the initial protein abundance and RPF is able to explain a large portion of the observed mRNA-protein discordance during mouse OET.

**Discussion**
Using a highly sensitive and quantitative LC–MS/MS approach, we systematically determined the proteome of mouse oocytes and early embryos and investigated its relationship with the corresponding translatome and transcriptome. Our results revealed distinct reprogramming of gene expression during oocyte maturation and embryo development at three levels. The largest transitions of both transcriptome and translatome occurred at the 2C stage (Fig. 2C), while the greatest global proteome change took place later, between the 8C and the blastocyst stage (Fig. 2C). Moreover, the proteome showed much fewer dynamically regulated genes (52.2%) than the translatome (83.5%) and the transcriptome (80.8%) during OET (Fig. 2D and Fig. 7E). One major reason for

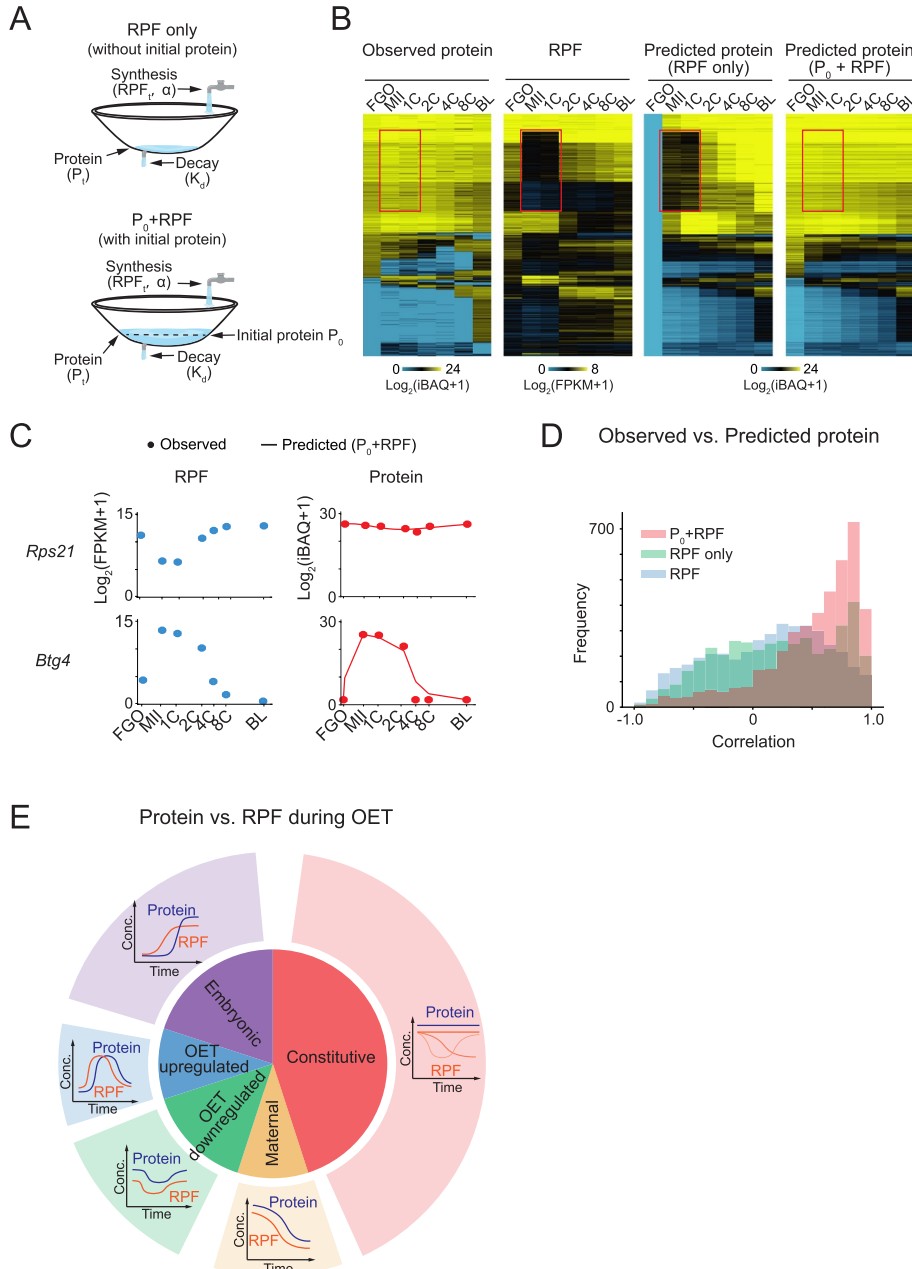

**Fig. 7** Protein dynamics could be predicted by RPF dynamics plus the initial protein abundance. **A** The kinetic models used for protein dynamics prediction. In the RPF-only model, protein abundance is determined by protein synthesis and degradation. In the $P_0 +$ RPF model, the initial protein level ($P_0$) is additionally considered. α, constant from translation to protein; $k_d$, degradation rate; $RPF_t$: translation level at time t; $P_t$: protein abundance at time t. **B** Heat maps showing the observed protein and RPF levels with the corresponding predicted protein dynamics using RPF-only and $P_0 +$ RPF models mapped. Red boxes indicate stages that show differential signals between observed protein and RPF. **C** Dot-line plots showing the protein levels predicted by the $P_0 +$ RPF model (line) and observed from LC–MS/MS (dot) for representative genes (right). RPF level at each stage is also shown (left). **D** Histogram showing the Spearman correlation coefficients between observed protein abundance and RPF level (blue) or predicted protein levels by RPF-only (green) and $P_0 +$RPF models (red). **E** Summary model of the protein and RPF dynamics in mouse oocytes and early embryos

the discrepancy between proteome and transcriptome/translatome is the dominance of FGO-stockpiled proteins in the proteomes of early embryos. Remarkably, among 103 strictly defined FGO-originated proteins (with minimal translation after fertilization), 58.3% could still be detected in blastocyst (Fig. 4C). These data underscore the importance of directly determining protein levels rather than relying on RNA or translation levels when studying oocytes and early embryos. Using a kinetic model, we showed that protein levels during mouse OET could be partially predicted with the initial protein abundance and translation profiles. Of note, the presence of stable oocyte proteins in OET appears to be an evolutionarily conserved feature as it was also observed in *Xenopus* [6].

One interesting question is why proteins in oocytes and early embryos are so stable. One possible reason is protein production is expensive for cells. The amino acid synthesis and polypeptide assembly could consume 50% of ATPs in rapidly growing yeast cells [42]. Therefore, stabilizing proteins may be especially efficient and energy conserving considering oocytes are often arrested at the diplotene stage even for years. It was reported that certain proteins are synthesized only once during the oocyte growth and execute important functions in the embryo [43, 44], underscoring the need for stability of these proteins. For example, CENP-A at mouse oocyte centromeres has been shown to persist over a year, which underpins the transgenerational inheritance of centromere identity through the female germline [43]. In addition, considering the sheer sizes of oocytes and embryos relative to somatic cells, the long-lasting oocyte-stockpiled proteins may be especially beneficial for embryos to save precious transcription resources for embryo-specific transcripts [45].

By contrast, the translatome and transcriptome experienced significant changes during oocyte maturation and early development. Another intriguing question is whether all such dynamic transcription and translation are functional given some of them do not appear to profoundly impact the proteome. This is particularly relevant to a group of genes that are translated in FGOs but stop translation upon meiotic resumption. Such dynamic translation contrasts the remarkably stable protein levels (Fig. 3A and 7E, protein constitutive-RPF OET downregulated). We speculate that the downregulation of translation and mRNAs may partially arise from the global deadenylation upon meiotic resumption [12, 46–48] as part of the cellular efforts to conserve resources especially when transcription is silent. In addition, the metabolism of nucleotides was reported to be increased during oocyte maturation [49]. The resulting nucleotide metabolites, such as purine and pyrimidine, may provide materials for later DNA and RNA synthesis during ZGA.

Our data revealed the protein variance in mouse oocytes and early embryos could be partially (43~80%) explained by translation and transcription changes (Fig. 5B). However, about 20~57% could not be explained by transcription and translation regulation (Fig. 5B). Similarly, it was reported only ~40% of differences in protein levels were attributed to variation in mRNA expression in mouse NIH3T3 cells [36]. Of note, protein levels can be regulated at the post-translational levels [50, 51]. How it plays a role in OET remains to be investigated in the future. Finally, it is worth noting that due to the limitation of detection sensitivity of LC–MS/MS, lowly expressed proteins were not analyzed here. New low-input proteomic technologies with improved sensitivity will be

instrumental to study low-abundance proteins, such as transcription factors, to further elucidate the regulation of OET.

## Conclusions

In this study, through integrative analyses of datasets generated by ultrasensitive mass spectrometry, we reveal distinct dynamics of the proteome compared to the translatome and the transcriptome during mouse OET. Specifically, for an extended period of oocyte maturation and early embryonic development, the proteomes are dominated by FGO-stockpiled proteins, while the transcriptome and translatome exhibit dynamic changes. Therefore, transcription and translation cannot simply be used to infer the protein levels during OET when post-transcriptional regulation is prevalent. We propose that the remarkably stable oocyte-originated proteomes may help conserve resources to accommodate the demanding needs of fast-growing embryos. These results not only illuminate the regulation of mammalian OET at different levels but also shed light on the fundamental principles controlling gene expression.

## Methods

### Animal maintenance

All animal maintenance and experimental procedures used in this study were carried out according to the guidelines of the Institutional Animal Care and Use Committee (IACUC) of Tsinghua University, Beijing, China. C57BL/6 J and PWK/PhJ mice were purchased from Vital River and Jackson Laboratory respectively and raised in a local core facility.

### Oocyte and early embryo collection

Pre-implantation embryos were collected from 4-week-old C57BL/6J female mice mated with PWK/PhJ males. To induce ovulation, the female mice were intraperitoneally injected with pregnant mare's serum gonadotropin (PMSG, 5 IU) and human chorionic gonadotrophin (hCG, 5 IU). Fully grown oocytes (FGOs) ($>70$ μm) were collected from the ovaries of C57BL/6J female mice 48 h after PMSG injection. MII oocytes or pre-implantation embryos were collected at the following time points after hCG stimulation: MII oocytes (14–16 h), 1-cell embryos (27–29 h, PN5), 2-cell embryos (46–48 h, late 2-cell stage), 4-cell embryos (54–56 h), 8-cell embryos (62–65 h). For blastocysts, embryos were collected at the 8-cell stage and cultured to blastocysts in vitro in KSOM medium (Millipore, MR-121-D). Oocytes and embryos were collected in the M2 medium (Sigma, M7167). The zona pellucida was gently removed by treatment with acidic Tyrode's solution (Sigma, T1788).

### Cycloheximide (CHX) treatment of mouse MII oocytes

To validate the quantification capability of LC–MS/MS, MII oocytes were collected and cultured in M2 medium containing either 0.1% DMSO or 100 μg/ml CHX (Sigma, C4859) for 24 h or 48 h.

**Sample preparation for LC–MS/MS**

For the proteome quantification, 100 oocytes or embryos of each developmental stage were harvested with two replicates. In addition, 10, 100, 200, and 500 FGOs were collected for the LC–MS/MS quantification ability validation. The oocytes or embryos were deprived of the zona pellucida in warm acidic Tyrode solution and washed with 1X PBS three times. Oocytes/embryos were then resuspended in lysis buffer (1% sodium deoxycholate, 10 mM TCEP, and 40 mM 2-chloroacetamide in 20 mM Tris–HCl, pH 8.5), boiled for 5 min, and sonicated to denature proteins, shear DNA and enhance cell disruption. Proteins were digested by trypsin and LysC with a ratio of 1:100 (enzyme:protein) and desalted by SDB-RPS StageTips. The eluted peptides were dried in SpeedVac and resuspended in 0.1% formic acid for analysis by mass spectrometry.

**LC–MS/MS analysis**

An UltiMateTM 3000 RSLCnano system, directly interfaced with a Q Exactive HF-X mass spectrometer was used here for LC–MS/MS analysis. Peptides were loaded to a trap column (75 µm × 20 mm, 3 µm C18,100 Å, 164,535, Thermo Fisher Scientific) with a max pressure of 620 bar using mobile phase A (0.1% formic acid in H2O), then separated on an analytical column (samples of FGOs to 8-cell embryos used 75 µm × 500 mm, 3 µm C18,100 Å, 164,570, Thermo Fisher Scientific; samples of blastocysts used 100 µm inner diameter, packed in house with ReproSil-Pur C18-AQ 1.9 µm resin from Dr. Maisch GmbH) with a gradient of 4–60% mobile phase B (80% acetonitrile and 0.08% formic acid) at a flow rate of 250 nl/min for 280 min. The MS analysis was operated in data-dependent acquisition (DDA) mode, with one full scan (300–1800 m/z, $R = 60,000$ at 200 m/z) at automatic gain control (AGC) of 3e6, followed by top 40 MS/MS scans with high energy collision dissociation (AGC of 1e5, maximum injection time (IT) 100 ms, isolation window 1.6 m/z, normalized collision energy of 27%).

**Processing of raw LC–MS/MS data**

Raw LC–MS/MS data of the proteome were processed by MaxQuant [52] software (v 1.6.2.10) for protein identification searching against the mouse UniProtKB database (September 2019 release). The search criteria were as follows: full tryptic specificity was required; two missed cleavages were allowed; carbamidomethylation (C) was set as the fixed modifications; oxidation (M) and acetylation (protein N terminal) were set as the variable modifications; ion mass tolerances were set at 10 ppm for all MS acquired in an Orbitrap mass analyzer. The false discovery rate was set to 0.01 for proteins and peptides and determined by searching a reverse database. Protein abundance was quantified using iBAQ intensity and log (base 2) transformed.

**LC–MS/MS protein intensity pre-processing**

The LC–MS/MS data in this study were performed in three batches (see below table).

| Batch | Sample |
| --- | --- |
| 1 | FGO_rep1, MII_rep1, 1C_rep1, 2C_rep1, 4C_rep1, 8C_rep1 |
| 2 | FGO_rep2, MII_rep2, 1C_rep2, 2C_rep2, 4C_rep2, 8C_rep2 |
| 3 | FGO_rep3, FGO_rep4, BL_rep1, BL_rep2 |

100 FGOs (1–2 replicates) were included in every batch to correct batch effects. Specifically, batch 1 was selected as the reference. FGO proteins detected in all three batches were used, and the differences of median intensities (across all proteins) in FGOs between batch 2/3 and batch 1 were computed as the batch correction factor. The intensities of proteins for all samples in the same batch were then corrected by subtracting the corresponding batch correction factor.

Next, as there are four replicates of FGOs, we averaged two replicates from batch 1 and batch 2 as the new replicate 1 and averaged the two replicates in batch 3 as the new replicate 2. FGO proteins that were only detected in one replicate were excluded. 27 genes that were not detected at any stages in the translatome or the transcriptome data were also discarded, leading to 4,108 genes for the downstream analysis. Wherever appropriate, average intensities were calculated using replicates in which the proteins were detected.

### RNA-seq and Ribo-lite library preparation and sequencing

For blastocysts, RNA-seq libraries were generated using the Smart-seq2 protocol as described previously [53]. Ribo-lite libraries were generated using the protocol described previously [12]. The RNA-seq and Ribo-seq data of all other stages were generated in our previous work [12].

### RNA-seq data processing

Raw reads were trimmed by Trim Galore v0.4.2 and then mapped to the transcriptome of mm9 by STAR v2.5.3a [54] with parameters –outFilterMultimapNmax 20 –outSAMstrandField intronMotif. The gene expression level was calculated by Cufflinks v2.2.1 [55] based on the annotation of mm9 from the UCSC genome browser. The average FPKM from two replicates was calculated.

### Ribo-seq data processing

Raw reads were trimmed by cutadapt v1.14 and then mapped to mouse rRNA sequences (mm9) using Bowtie2 v2.2.2 [56] with parameters –seed = 23. Those aligned to rRNA were discarded, and the rest reads were mapped to the transcriptome of mm9 using STAR v2.5.3a [54] with parameters –outFilterMismatchNmax 2 –outFilterMultimapNmax 20 –outFilterMatchNmin 16 –alignEndsType EndToEnd. The gene expression levels were then calculated by Cufflinks v2.2.1 [55] based on the annotation of the CDS regions, defined by the mm9 refFlat database from the UCSC genome browser. We next calculated the average FPKM for replicates.

### Clustering analysis

The K-means clustering of LC–MS/MS signal was conducted using Cluster 3.0 [57] with the parameters -g 7 (Euclidean distance) -k 10 -r 100. The embryonic proteins detectable at different stages after ZGA were merged into one cluster. Heat maps were generated using Java Treeview.

**Identification of dynamically and stably expressed genes for the proteome, translatome, and transcriptome**

To estimate the global dynamic extents for proteome, translatome, and transcriptome during OET, the coefficient of variance (CV) analysis was applied to estimate the overall variance of each gene across developmental stages. For mRNAs and RPFs, genes that expressed (FPKM > = 1) at one or more stages were selected. Genes with CV > 0.2 of log2 transformed intensity of protein, RPF, or mRNA were considered as dynamically expressed across stages. The rest genes were regarded as stably expressed.

**Analyses of parent-of-origin dynamics of proteins**

The single nucleotide polymorphisms (SNPs) between the C57BL/6N and PWK/PhJ mouse strains were downloaded from Sanger database (https://www.mousegenomes.org/snps-indels/) and annotated using ANNOVAR [58] based on the mm9 genome assembly. A customized protein allele database was constructed using ANNOVAR with the parameters "annotate_variation.pl -buildver mm9 -geneanno –seq_padding 30 -dbtype knowngene". The mouse UniProt protein sequences were updated by replacing the specific amino acids of nonsynonymous variants. For each nonsynonymous variant, peptides of 61 amino acids were generated, comprising 30 amino acids upstream and downstream of the variant residue. Subsequently, the LC–MS/MS data were searched against this allele peptide database using MaxQuant [52], with a false discovery rate of 0.01 for peptide detection.

The intensity of allele peptides from different batches was processed in the same manner as protein intensity to eliminate batch effects. The intensities of different allele peptides from the same proteins were averaged. To detect proteins exhibiting allele-specific expression (ASE), a t-test was performed between alleles with a *P-value* < 0.05 and a |fold change| greater than 2. The proteins exhibiting ASE at any stage of early embryos were identified.

**Analyses of parent-of-origin dynamics of RNAs**

The paired-end reads of RNA-seq were first aligned to a modified mm9 genome where all polymorphic sites were N-masked, using hisat2 [59]. After that, the polymorphic sites were identified on the aligned reads. Reads that did not contain SNP information or contained conflicting allelic polymorphic sites were classified as unassigned. Read pairs in which both reads were assigned to the same parental allele or one read was assigned to one parental allele and the other was unassigned were classified as allelic reads for downstream analysis. The abundance of RNA alleles was quantified by StringTie [60] using the refFlat database from the UCSC genome browser.

**DEP and DEG analyses**

The differentially expressed proteins (DEPs) between two consecutive stages based on LC–MS/MS and differentially expressed genes (DEGs) based on RNA-seq and Ribo-seq were identified by twofold change and Student's t-test *P-value* < 0.05. The DEPs for CHX treatment of MII oocytes were defined by a fourfold change.

### Prediction of intrinsically disordered regions (IDRs)

All validated open reading frames (ORFs) in the mouse genome were downloaded from UniProtKB/SwissProt (September, 2019 release). The intrinsic disorder regions were predicted for all protein sequences using the web tool IUPred2A based on the energy estimation [61]. We analyzed the lengths and locations of disordered segments in the protein sequences with an in-house script. Residues with a prediction score above 0.5 were considered as disordered and only regions that contained more than 30 continuous disordered residues were set as IDRs. This length cut-off for IDR was selected based on previous studies [62, 63] showing that there is a minimum length of about 30 residues that allows a disordered protein terminus to efficiently initiate degradation [64].

### Quantification and statistical analysis

The reproducibility between replicates and correlations among protein and RPF or mRNA were estimated with Spearman rank correlation. All box and violin plots were plotted using R and Python. Statistical significance for the enrichment of dormant RNA in CHX repressed proteins was assessed with Fisher's Exact test. The significance of the association between protein-RPF/mRNA correlation and protein FGO abundance, protein half-life time, and IDR was estimated with Wilcoxon rank-sum test (two-tailed).

### Gene ontology analysis

The Metascape web tool [65] was used to identify the enriched Gene Ontology terms. The terms for each functional cluster were summarized to a representative term and P-values were plotted to show the significance.

### Prediction of protein dynamics

Following a previous study [6], we built a kinetic model to predict protein dynamics across developmental stages based on the law of mass action. In brief, assuming the spatially and temporally constant rates of synthesis and degradation, the expected change in protein level over time is given by

$$\frac{dp(t)}{dt} = \alpha r(t) - k_d p(t),$$

where p(t) is the abundance of proteins at time t, r(t) is the abundance of translating mRNAs (i.e., RPF), $\alpha$ is the transition rate from mRNA translation to protein, and $k_d$ is the degradation rate of protein. For each protein, the parameters $\alpha$ and $k_d$ can be inferred by giving the measurements of r(t) and p(t) and the initial protein abundance $p_0$. Mathematically, we aimed to minimize the difference between the observed protein level $\widehat{p}_i$ at time $t_i$ and the predicted protein level $p(t_i | p_0, \alpha, k_d)$ on average over all observed time points i, which can be formulated as

$$\min_{\{\alpha, k_p\} \geq 0} \sum_i \left( p(t_i | p_0, \alpha, k_d) - \widehat{p}_i \right)^2.$$

We determined the parameters $\alpha$ and $k_d$ in the above optimization problem by the function fmincon in MATLAB using the respective RPF and protein abundance data and initial protein abundance $p_0$, which was termed as $P_0 + \text{RPF}$ model. To assess the effects of $p_0$ on protein prediction, we set $p_0$ as 0 when solving the above optimization problem, which was termed as RPF only model. Of note, because RPF data were only measured at specific time points, we thus utilized the piecewise interpolation to approach $r(t)$. The abundance of RPF at any time t $(I_h(t))$ is given by

$$I_h(t) = \frac{t - t_{k+1}}{t_k - t_{k+1}} r(t_k) + \frac{t - t_k}{t_{k+1} - t_k} r(t_{k+1}), t \in [t_k, t_{k+1}], k = 1, 2, \ldots, n-1,$$

where n is the number of time points with RPF data.

## Supplementary Information

---

**Additional file 1: Supplementary figure and figure legends. Figure S1.** Validation of LC-MS/MS quantification ability in mouse oocytes. **Figure S2.** Evaluation of LC-MS/MS data quality in mouse oocytes and early embryos. **Figure S3.** Dynamic changes of RPF and mRNA in the mouse oocytes and early embryos. **Figure S4.** Correlation between protein and mRNA across developmental stages. **Figure S5.** Differentially expressed proteins and genes at each consecutive stage. **Figure S6.** Correlation between protein and RPF or mRNA for individual genes across developmental stages. **Figure S7.** Protein features related to the concordance between protein and RPF or mRNA.

**Additional file 2: Table S1.** The processed proteomics data in this study.

**Additional file 3: Table S2.** The high-confidence proteomics data and the matched Ribo-seq and RNA-seq data.

**Additional file 4: Table S3.** Parent-of-origin dynamics of oocyte-originated and embryonic proteins. Related to Figure 4D.

**Additional file 5: Table S4.** Defined up- and down-regulated proteins, RPFs, and mRNAs between each of the two consecutive stages. Related to Figure 5A.

**Additional file 6.** Review history

---

### Acknowledgements
We are grateful to members of the Xie laboratory for discussion and comments during the preparation of the manuscript, and to the Animal Center, Proteomics Center, and Biocomputing Facility at Tsinghua University for their support.

### Peer review information

### Review history
The review history is available as Additional file 6.

### Authors' contributions
S.J. and W.X. conceived and designed the project. S.J., J.M., F.K., L.W., and B.H. collected the mouse oocytes and embryos. L.W. and K.Z. performed the CHX treatments on MII oocytes. Y.C. performed the LC–MS/MS and quantification of protein intensity. Z.Z. performed the RNA-seq and Ribo-seq for blastocyst. H.Z. and K.Z. performed the data analysis. S.W. provided the code of protein dynamics prediction. Z. X., K. X., and Z. L. provided the Ribo-seq and RNA-seq data of FGO to 8C embryos. H.Z., K.Z., and W.X. prepared most figures. H.Z., S.J., K.Z., and W. X. wrote the manuscript with help from all authors.

### Funding
National Key R&D Program of China, 2022YFC2702300.
National Natural Science Foundation of China 31,988,101 and 31,725,018.
National Key R&D Program of China 2019YFA0508900.
Beijing Municipal Science and Technology Commission Z181100001318006.
Tsinghua-Peking Center for Life Sciences.
Wei Xie is a recipient of HHMI International Research Scholar award and New Cornerstone Investigator award.

### Availability of data and materials
The mass spectrometry data generated for this study have been deposited to the ProteomeXchange Consortium via PRIDE [66] with the accession number PXD035696 [67]. The RNA-seq and Ribo-seq data of blastocysts are deposited to Gene Expression Omnibus (GEO) with the accession number GSE209648 [68]. Previously published Ribo-seq and RNA-seq data [12, 25] used in this work were from NCBI GEO accession number GSE165782 (Ribo-seq, FGO, MII oocytes, 1C, 2C, 4C, and 8C embryos; RNA-seq, FGO, MII oocytes) [69] and GSE71434 (RNA-seq, 1C, 2C, 4C, and 8C embryos) [70].

The published protein profile of mouse oocytes and/or early embryos from Wang et al. [15], Gao et al. [16], and Israel et al. [17] was downloaded directly from the processed data provided by each study. The protein half-life datasets were obtained from the processed data of mouse embryonic neurons [37] and mouse embryonic fibroblast cell line NIH3T3 [36].

## Declarations

### Ethics approval and consent to participate
All animal protocols in this study were approved by the Ethics Committee of Tsinghua University (number: 17-XW1).

### Competing interests
The authors declare no competing interests.

### Author details
[1]Center for Stem Cell Biology and Regenerative Medicine, MOE Key Laboratory of Bioinformatics, New Cornerstone Science Laboratory, School of Life Sciences, Tsinghua University, Beijing 100084, China. [2]Tsinghua-Peking Center for Life Sciences, Beijing, China. [3]School of Life Sciences, Tsinghua University, Beijing, China. [4]School of Mathematics and Statistics, Wuhan University, Wuhan, China. [5]Hubei Key Laboratory of Computational Science, Wuhan University, Wuhan, China. [6]Academy for Advanced Interdisciplinary Studies, Peking University, Beijing 100871, China. [7]Zhejiang Provincial Key Laboratory of Pancreatic Disease, School of Medicine, the First Affiliated Hospital, Zhejiang University, Hangzhou 310002, China.

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

## 

