## [**Additional file 6.** Review history · Genome Biology]

Review History

First round of review

Reviewer 1

Are you able to assess all statistics in the manuscript, including the appropriateness of statistical tests used? Yes, and I have assessed the statistics in my report.

Comments to author:

Cells experience extensive reprogramming during the OET process, during which transcription is silent before ZGA. Therefore, mechanisms beyond transcription are essential for this process. In this study, the authors characterized the most comprehensive proteome for seven stages from FGO to blastocyst, analyzed its relationship with mRNA and ribosome engaged mRNA, and present a good kinetic model that can well predict protein dynamics across the OET. The findings here are very interesting and informative. In addition, the proteome data will be an extremely valuable resource for understanding OET. The manuscript is very well written and is a very good candidate for Genome Biology. I have several minor comments for further improvement of the manuscript.

1. The embryos used are from B6 females mated with PWK males. It will be very interesting to analyze the parent-of-origin of protein detected in the embryos if possible. I understand that the coverage might be an issue for this kind of analysis. Therefore, it is a suggestion but not a requirement.
2. The data in Figure 1a nicely demonstrated the MS technology in quantifying the proteome from 10 oocytes to 500 oocytes. How the genes are selected in the scatter plots is not described. In addition, it is better to describe how many proteins are detected from different number of input oocytes, which can give the readers better feeling of how much sample is needed for their purposes.
3. It is better to include the list of genes/proteins in a supplemental table for Figures, such as 3A and 4E, to facilitate the readers to better use the dataset.
4. The correlation between protein and RPF for the most highly expressed proteins looks to be significant negative correlation. A discussion about this observation will be a plus.
5. "with 600-8,000 embryos at each stage" shall be "with 600-8,000 embryos or oocytes at each stage".
6. "TUBLIN" in Figure S1D shall be "TUBULIN".
7. The previous knowledge of proteins when describing examples of dynamics of individual proteins shall be properly referenced.
8. "5) Is "Embryonic" proteins are translated upon ZGA" better to be described as "5) "Embryonic" proteins are translated after ZGA"?

Reviewer 2

Are you able to assess all statistics in the manuscript, including the appropriateness of statistical tests used? No, I do not feel adequately qualified to assess the statistics.

Comments to author:

This study provides the LC-MS/MS data (iBAQ) to match the previously published Ribo-seq and RNA-seq datasets. It aims to illuminate the oocyte-to-embryo transition by comparing and contrasting the detected proteins (current data), with the translome and the transcriptome (previous data). Overall, it is a solid, valuable and overdue study. It will be of interest to developmental biologists with a focus on embryology. Strengths of the study are the sensitivity (samples contained only 100 oocytes or embryos), the broad coverage (about 4100 proteins), the multi-omics integrative analysis, and the comparison with other studies. I cannot think of substantial weaknesses. Perhaps the part of the study involving the half-lives of the proteins is not as strong as the other parts, but this is because there are no datasets available on the stability of proteins in mammalian oocytes and embryos. I embrace the study's conclusion that "the remarkably stable oocyte-originated protein may help save precious transcription resources for zygotic genes to accommodate the demanding needs of growing embryos": this makes a lot of sense to me. An additional possibility is related to maternal aging (PMID 27040782) i.e. some proteins are synthesized only once per life time, and therefore must be stable. This additional possibility might be something to consider in the Discussion.

I have the following remarks.

The zona pellucida - the extracellular coat of the oocyte and preimplantation embryo - was removed using Tyrode solution, which was a wise thing to do, in order to use the sensitivity of the mass spectrometer for the intracellular proteins. Why then were ZP1, ZP2 and ZP3 still detected as proteins even at late preimplantation stages?

It appears that the embryos up to 8-cell stage were recovered *in vivo*, and the blastocysts were produced *in vitro* from culture of 8-cell embryos in KSOM. Here the Authors should consider that cleavage in oviducts of mice stimulated with gonadotropins can be detrimental to embryos (this has been reported countless times, and most recently summarized here: PMID 34887460). It should be mentioned in the manuscript that the effect of gonadotropins on embryo quality and thereby on the OET may have influenced the results of the study.

It appears (Fig. 3A, Fig.6B) that very few of the observed proteins are degraded during preimplantation development (color code going from yellow to blue). By contrast, many more proteins seem to appear *de novo* (color code going from blue to yellow). Could the Authors please elaborate on this observation i.e. on why there are more proteins appearing (putative embryonic) than proteins disappearing (putative maternal).

It appears that some popular transcription factors known to be present in oocyte and preimplantation embryos, like Oct4/Pou5f1, were not detected in the present study. It is odd that Cdx2 is detected, but Oct4 is not. Clearly, reducing the amount of input material is good for many reasons (e.g. fewer animals killed), however, it goes perhaps at the expense of proteome

depth. Apart from transcription factors, I note that also an abundant housekeeping gene product (HPRT) was not detected either. Could the Authors please elaborate on this issue. I am not trying to criticize the Authors, I am only trying to understand these phenomena, which I encountered also in my own mass spec studies of early embryos. Is it purely stochastic, or there are perhaps technical factors involved in accounting for proteins that are there but go undetected?

How to explain that Npm1 and Npm2, which are sperm-specific, were detected throughout preimplantation?

I thank the Authors and the Journal for sharing with me this very nice piece of work.

AUTHORS' RESPONSES TO REVIEW

GENERAL RESPONSES

We deeply appreciate the reviewers for their valuable comments and suggestions, which have led to significant improvement of the manuscript. Here we include a letter to address the issues raised in the previous submission. Please note to avoid confusion, we used Fig. 1, 2, 3, etc. to refer to figures in the manuscript and Fig. R1, R2, R3, etc. to refer to figures in this letter.

Please note that for the reviewers' convenience, we also included a version of the manuscript in which the revised sections were highlighted related to our responses to the reviewers' comments.

Response to the Reviewers:

Reviewer #1:

Cells experience extensive reprogramming during the OET process, during which transcription is silent before ZGA. Therefore, mechanisms beyond transcription are essential for this process. In this study, the authors characterized the most comprehensive proteome for seven stages from FGO to blastocyst, analyzed its relationship with mRNA and ribosome engaged mRNA, and present a good kinetic model that can well predict protein dynamics across the OET. The findings here are very interesting and informative. In addition, the proteome data will be an extremely valuable resource for understanding OET. The manuscript is very well written and is a very good candidate for Genome Biology. I have several minor comments for further improvement of the manuscript.

Response: We sincerely thank the reviewer for these encouraging comments. We have now carefully addressed these questions following the reviewer's suggestions, which we believe have substantially helped improve this manuscript.

Comment 1.1. *The embryos used are from B6 females mated with PWK males. It will be very interesting to analyze the parent-of-origin of the protein detected in the embryos if possible. I understand that the coverage might be an issue for this kind of analysis. Therefore, it is a suggestion but not a requirement.*

Response: This is an excellent comment and we thank the reviewer for bringing this analysis to our attention. We have now performed the parent-of-origin analysis for proteins. A total of 484 proteins with non-synonymous amino acid variations caused by SNPs were identified by LC-MS/MS, among which 173 variant proteins showed differential expression between the two alleles at one or more developmental stages in early embryos (Fig. R1a). Among them, 117 proteins were identified as likely maternal proteins that originated from the oocyte and persisted in early embryos with no paternal expression detected (Fig. R1a-b, “oocyte-specific”), including known maternal factors such as *Nlrp5*, *Zp2*, *Gdf9*, *Dppa3*, and *Plat*. Consistently, their paternal RNAs were generally low in early embryos, suggesting the absence of zygotic transcription. Oocyte-originated transcript levels also reduced abruptly after ZGA. This contrasts with the stable proteins, again supporting the notion that maternal proteins are degraded slower than mRNAs in early embryos. In addition, 56 proteins displayed oocyte expression and persisted until the embryonic stage, with paternally-derived proteins also detected at certain stages after ZGA, mostly at the blastocyst stage (Fig. R1a-b, “oocyte-embryonic”). In agreement with the protein dynamics, transcripts of maternal origin were present throughout oocytes and early embryos, and transcripts of paternal origin emerged after ZGA. We also identified 21 and 4 proteins that were only detected in embryos and showed exclusively maternal and paternal origin, respectively (Fig. R1a, “Embryonic-maternal” and “Embryonic-paternal”). However, the majority of them showed biallelic mRNA expression in embryos, raising the possibility that they are likely false positives due to detection dropout in LC-MS/MS. Overall, these results indicate a generally concordant relationship between transcript and protein dynamics in mouse oocytes and early embryos, supporting the slow degradation of oocyte-derived proteins in mouse embryos. We have now added these results to the revised manuscript (pages 8-9, lines 177-195; Fig. 4D-E; Methods, lines 463-486), and provided the full data of parent-of-origin protein expression in Table S3.

Fig. R1. Analyses of parent-of-origin expression of proteins.

(a) Heat maps showing the parent-of-origin dynamics of proteins in oocytes and early embryos, with the corresponding parent-of-origin mRNA levels mapped. The example genes are also listed. n, gene number.

(b) Line plots showing the parent-of-origin dynamics of proteins and mRNAs across developmental stages for example genes.

Comment 1.2. The data in Figure 1a nicely demonstrated the MS technology in quantifying the proteome from 10 oocytes to 500 oocytes. How the genes are selected in the scatter plots is not described. In addition, it is better to describe how many proteins are detected from the different numbers of input oocytes, which can give the readers a better feeling of how much sample is needed for their purposes.

Response: We apologize for missing this information. We have now included this information in Fig. R2 below and in the revised manuscript (page 4, lines 46-48, Fig. S1B).

“A total of 1,896, 3,072, 3,288, and 4,185 proteins were identified from 10, 100, 200, and 500 FGOs, respectively (Fig. S1B). Proteins consistently detected in all FGO samples (n=1,813) were then used to evaluate the quantification capability of LC-MS/MS”.

Fig. R2. The number of proteins detected with varying numbers of FGOs. n, oocyte number. FGOs, full-grown oocytes.

Comment 1.3. It is better to include the list of genes/proteins in a supplemental table for Figures, such as 3A and 4E, to facilitate the readers to better use the dataset.

Response: We thank the reviewer for the great suggestion. We have included the complete tables for Fig. 3A and Fig. 4E as Table S2 and Table S4, respectively.

Comment 1.4. The correlation between protein and RPF for the most highly expressed proteins looks to be a significant negative correlation. A discussion about this observation will be a plus.

Response: We thank the reviewer for the question. We assume the “highly expressed proteins” referred to the constitutive proteins that are abundant in FGOs and maintain high levels throughout early development. These proteins were accumulated in large quantities during oocyte growth and remained stable due to their long half-lives (Fig. R3a). Moreover, we noted that the RPFs and mRNAs for a significant proportion of constitutive proteins were downregulated from MII oocytes to early embryos (Fig. R3b, 2nd and 3rd groups), while new proteins were continuously produced, albeit at a growingly slower speed. As a result, the correlation between RPF/mRNA and protein even becomes negative (Fig. R3c-d). We have now added the related discussion in the revised manuscript (page 11, lines 264-269; Fig. S7B and C).

Fig. R3. Correlation between protein, RPF, and mRNA dynamics for constitutive proteins.

(a) Box plots showing half-lives determined in mouse embryo neuron (Mathieson et al., 2018) or embryonic fibroblast cell line NIH3T3 (Schwanhausser et al., 2011) for different protein groups. The significance for all plots was calculated by the Wilcoxon rank-sum test (two-tailed). ***, P -value < 0.001; **, P -value < 0.01.

(b) Heat maps showing the K-means clustering results based on protein dynamics in oocytes and early embryos ($n = 4,108$), with the corresponding RPF and mRNA levels mapped. Constitutive proteins are further classified into four subgroups based on the dynamics of RPFs. 1st subgroup, protein constitutive-RPF constitutive; 2nd subgroup, protein constitutive-RPF OET downregulated; 3rd subgroup, protein constitutive-RPF low; 4th subgroup, protein constitutive-RPF maternal.

(c) Spearman correlation between protein and RPF for subgroups of constitutive proteins. n indicates protein number.

(d) Spearman correlation between protein and mRNA for subgroups of constitutive proteins. n indicates protein number.

Comment 1.5. "with 600-8,000 embryos at each stage" shall be "with 600-8,000 embryos or oocytes at each stage".

Response: We thank the reviewer for pointing this out. We have now revised it in the manuscript (page 3, line 24).

Comment 1.6. "TUBLIN" in Figure S1D shall be "TUBULIN".

Response: We thank the reviewer for pointing this out. This is now revised (Fig. S1E in the revised manuscript).

Comment 1.7. The previous knowledge of proteins when describing examples of the dynamics of individual proteins shall be properly referenced.

Response: We thank the reviewer for the suggestion. We have now carefully gone through the manuscript and added proper references for individual proteins or genes as follows (page 7, lines 138-142):

“The upregulation of TET3 upon meiotic resumption is consistent with its role in the upcoming global DNA methylome demethylation (Gu et al., 2011) and also explains why it does not demethylate the oocyte genome”.

“The upregulation of EED and EZH2, components for Polycomb repressive complex 2 (PRC2), is interesting as oocyte-inherited H3K27me3 enables allele-specific expression of key imprinted genes (e.g. *Gab1*, *Sfmbt2*, and *Slc38a4*) and X chromosome genes (*Xist*) in early embryos (Harris et al., 2019; Inoue et al., 2018; Inoue et al., 2017)”.

Comment 1.8. "5) Is "Embryonic" proteins are translated upon ZGA" better to be described as "5) "Embryonic" proteins are translated after ZGA"?

Response: We thank the reviewer for the suggestion. We have now revised it in the manuscript (page 7, line 143).

Reviewer #2:

This study provides the LC-MS/MS data (iBAQ) to match the previously published Ribo-seq and RNA-seq datasets. It aims to illuminate the oocyte-to-embryo transition by comparing and contrasting the detected proteins (current data), with the translome and the transcriptome (previous data). Overall, it is a solid, valuable and overdue study. It will be of interest to developmental biologists with a focus on embryology. Strengths of the study are the sensitivity (samples contained only 100 oocytes or embryos), the broad coverage (about 4100 proteins), the multi-omics integrative analysis, and the comparison with other studies. I cannot think of substantial weaknesses. Perhaps the part of the study involving the half-lives of the proteins is not

as strong as the other parts, but this is because there are no datasets available on the stability of proteins in mammalian oocytes and embryos.

Response: We sincerely thank the reviewer for the great summary of our study and the encouraging comments.

I have the following remarks.

Comment 2.1. *I embrace the study's conclusion that "the remarkably stable oocyte-originated protein may help save precious transcription resources for zygotic genes to accommodate the demanding needs of growing embryos": this makes a lot of sense to me. An additional possibility is related to maternal aging (PMID 27040782) i.e. some proteins are synthesized only once per life time, and therefore must be stable. This additional possibility might be something to consider in the Discussion.*

Response: This is an excellent point. We have now added a related discussion in the revised manuscript (page 13, lines 333-337) as follows:

“It was reported that certain proteins are synthesized only once during the oocyte maturation and execute important functions in the embryos (Burkhardt et al., 2016; Smoak et al., 2016), underscoring the need for stability of these proteins. For example, CENP-A at mouse oocyte centromeres has been shown to persist over a year, which underpins the transgenerational inheritance of centromere identity through the female germline (Smoak et al., 2016)”.

Comment 2.2. *The zona pellucida - the extracellular coat of the oocyte and preimplantation embryo - was removed using Tyrode solution, which was a wise thing to do, in order to use the sensitivity of the mass spectrometer for the intracellular proteins. Why then were ZP1, ZP2 and ZP3 still detected as proteins even at late preimplantation stages?*

Response: We thank the reviewer for pointing this question out. ZP1, ZP2, and ZP3 were indeed detected from oocytes to blastocysts in our LC-MS/MS data, except for ZP1 and ZP2 in the 8C embryos likely due to the detection dropout of LC-MS/MS (Gromski et al., 2014; Jin et al., 2018) (Fig. R4a). Consistently, ZP proteins were also detected in previous studies in zona pellucida removed oocytes and embryos by LC-MS/MS (Israel et al., 2019), Western blot (Taher et al., 2021), or immunofluorescence (Hoodbhoy et al., 2006) (Fig. R4b-d). This may reflect the existence of

either the ZP protein precursors as suggested (Taher et al., 2021), or the free ZP proteins outside of zona pellucida.

Fig. R4. Expression of ZP1/2/3 in the mouse oocytes and early embryos.

(a) ZP1/2/3 protein levels in oocytes and early embryos identified by LC-MS/MS in this study.

(b) ZP1/2/3 protein levels in oocytes and early embryos identified by LC-MS/MS in the study of Israel et al. (Israel et al., 2019).

(c) Western blots showing ZP3 protein in MII oocytes with zona pellucida removed from natural ovulation and superovulation and zona-intact MII oocytes from the superovulation (Taher et al., 2021).

(d) ZP2 and ZP3 immunofluorescence from Hoodbhoy et al. (Hoodbhoy et al., 2006). Growing mouse oocytes with zona pellucida removed were immunolabeled with fluorescently conjugated anti-ZP2 and anti-ZP3. Endogenous ZP2 (**A**) and ZP3 (**B**) colocalized (**C**) in the juxtannuclear endoplasmic reticulum (arrowhead), multivesicular aggregates (arrows), and plasma membrane are indicated.

Comment 2.3. It appears that the embryos up to the 8-cell stage were recovered in vivo, and the blastocysts were produced in vitro from the culture of 8-cell embryos in KSOM. Here the Authors should consider that cleavage in oviducts of mice stimulated with gonadotropins can be detrimental to embryos (this has been reported countless times, and most recently summarized here: PMID 34887460). It should be mentioned in the manuscript that the effect of gonadotropins on embryo quality and thereby on the OET may have influenced the results of the study.

Response: We thank the reviewer for the alert. We have now added this information in the revised manuscript (page 5, lines 77-80) as follows:

“We collected the oocytes and embryos *in vivo* up to the 8C stage and obtained blastocysts through *in vitro* culturing starting from the 8C embryos for the convenience of sample collection. It is worth noting that superovulation and *in vitro* culturing may influence the qualities of mouse embryos and their proteomes (Taher et al., 2021)”.

Comment 2.4. It appears (Fig. 3A, Fig.6B) that very few of the observed proteins are degraded during preimplantation development (color code going from yellow to blue). By contrast, many more proteins seem to appear de novo (color code going from blue to yellow). Could the Authors please elaborate on this observation i.e. why there are more proteins appearing (putative embryonic) than proteins disappearing (putative maternal)?

Response: We thank the reviewer for this question. We believe that this is because the FGO-originated proteins are often highly abundant and undergo slow degradation in early embryos, thus showing as “constitutive proteins”. This correlates with their long half-lives (Fig. R3a). A large part of FGO-originated proteins can persist to the blastocyst stage, despite their transcription and translation being repressed after the 2C stage (Fig. R5a, red box). For example, among 103 strictly defined FGO-originated proteins with no or low translation after fertilization (RPF FPKM<5 at the 1C stage and afterward), 58.3% were still detectable in blastocysts (Fig. R5b).

Fig. R5. Global proteome dynamics in mouse oocytes and pre-implantation embryos.

(a) Protein, RPF, and mRNA dynamics in oocytes and early embryos are shown. The red box indicates FGO-originated proteins that persist to blastocyst, even though their transcription and translation are already repressed after the 2C stage.

(b) Left, violin plot showing the protein levels of all proteins ($n = 4,108$). Blue dots are proteins that originated from FGOs but were not translated or lowly translated in early embryos (FGO originated, RPF FPKM < 5 at the 1C stage and afterward). Right, line plots show the protein, RPF, and mRNA dynamics of representative genes.

Comment 2.5. It appears that some popular transcription factors known to be present in oocyte and preimplantation embryos, like Oct4/Pou5f1, were not detected in the present study. It is odd that Cdx2 is detected, but Oct4 is not. Clearly, reducing the amount of input material is good for many reasons (e.g. fewer animals killed), however, it goes perhaps at the expense of proteome depth. Apart from transcription factors, I note that also an abundant housekeeping gene product (HPRT) was not detected either. Could the Authors please elaborate on this issue? I am not trying to criticize the Authors, I am only trying to understand these phenomena, which I encountered also

in my own mass spec studies of early embryos. Is it purely stochastic, or there are perhaps technical factors involved in accounting for proteins that are there but go undetected?

Response: We thank the reviewer for the careful observation. For HPRT1 and POU5F1, they were filtered out during data analysis as they were only detected in one of the two replicates of FGOs. As we used FGOs as a reference to correct batch effects (Method, lines 415-429), for proteins that do have expression in FGOs, we required them to be detected in both FGO replicates. The HPRT1 and POU5F1 proteins were indeed detected at other stages, such as the blastocyst. Their protein levels are relatively lower in FGOs compared to other stages (Fig. R6a), likely explaining why they were missed in one replicate. Our analysis showed that proteins variably identified in replicates of FGOs tended to have much lower abundance than proteins that were consistently detected in both replicates of FGOs (Fig. R6b). Therefore, more sensitive proteome technologies are warranted in the future to detect lowly expressed proteins. We have now provided the full proteome dataset in Table S1.

Fig. R6. Detection dropout of mass spectrometry in mouse oocytes and early embryos.

(a) HPRT1 and POU5F1 protein levels in oocytes and early embryos in two replicates.

(b) The abundance of FGO proteins that were detected in both ("Consistent") or only one replicate ("Variable") of FGOs.

Comment 2.6. How to explain that Npm1 and Npm2, which are sperm-specific, were detected throughout preimplantation?

Response: We appreciate the reviewer's comment. We believe that NPM1 and NPM2 are not sperm-specific proteins. Previous studies showed they are present in both oocytes and preimplantation embryos (Fig. R7a-b) (Burns et al., 2003; Inoue and Aoki, 2010; Zatsepina et al., 2003), consistent with our proteome data (Fig. R7c). It was shown that NPM2 and NPM1 play

critical roles in the assembly of nucleolus precursor bodies (NPBs) and embryonic development, respectively (Burns et al., 2003; Inoue and Aoki, 2010; Zatssepina et al., 2003).

Fig. R7. Expression of NPM1 and NPM2 in mouse oocytes and early embryos.

(a) Immunofluorescence analysis of endogenous NPM2 protein in oocytes, 1-cell, and 8-cell embryos (Burns et al., 2003). PMSG, pregnant mare serum gonadotropin.

(b) Subcellular localization of exogenous EGFP–NPM2 and endogenous NPM1 (B23) in oocytes and preimplantation embryos from Inoue et al. (Inoue and Aoki, 2010).

(c) Dynamics of NPM1 and NPM2 proteins in mouse oocytes and early embryos in this study.

I thank the Authors and the Journal for sharing with me this very nice piece of work.

Response: Thank you.

References:

Burkhardt, S., Borsos, M., Szydłowska, A., Godwin, J., Williams, S.A., Cohen, P.E., Hirota, T., Saitou, M., and Tachibana-Konwalski, K. (2016). Chromosome Cohesion Established by Rec8-Cohesin in Fetal Oocytes Is Maintained without Detectable Turnover in Oocytes Arrested for Months in Mice. *Curr Biol* 26, 678-685.

- Burns, K.H., Viveiros, M.M., Ren, Y., Wang, P., DeMayo, F.J., Frail, D.E., Eppig, J.J., and Matzuk, M.M. (2003). Roles of NPM2 in chromatin and nucleolar organization in oocytes and embryos. *Science* 300, 633-636.
- Gromski, P.S., Xu, Y., Kotze, H.L., Correa, E., Ellis, D.I., Armitage, E.G., Turner, M.L., and Goodacre, R. (2014). Influence of missing values substitutes on multivariate analysis of metabolomics data. *Metabolites* 4, 433-452.
- Hoodbhoy, T., Aviles, M., Baibakov, B., Epifano, O., Jimenez-Movilla, M., Gauthier, L., and Dean, J. (2006). ZP2 and ZP3 traffic independently within oocytes prior to assembly into the extracellular zona pellucida. *Mol Cell Biol* 26, 7991-7998.
- Inoue, A., and Aoki, F. (2010). Role of the nucleoplasmin 2 C-terminal domain in the formation of nucleolus-like bodies in mouse oocytes. *FASEB J* 24, 485-494.
- Israel, S., Ernst, M., Psathaki, O.E., Drexler, H.C.A., Casser, E., Suzuki, Y., Makalowski, W., Boiani, M., Fuellen, G., and Taher, L. (2019). An integrated genome-wide multi-omics analysis of gene expression dynamics in the preimplantation mouse embryo. *Sci Rep* 9, 13356.
- Jin, Z., Kang, J., and Yu, T. (2018). Missing value imputation for LC-MS metabolomics data by incorporating metabolic network and adduct ion relations. *Bioinformatics* 34, 1555-1561.
- Mathieson, T., Franken, H., Kosinski, J., Kurzawa, N., Zinn, N., Sweetman, G., Poeckel, D., Ratnu, V.S., Schramm, M., Becher, I., *et al.* (2018). Systematic analysis of protein turnover in primary cells. *Nat Commun* 9, 689.
- Schwanhaussner, B., Busse, D., Li, N., Dittmar, G., Schuchhardt, J., Wolf, J., Chen, W., and Selbach, M. (2011). Global quantification of mammalian gene expression control. *Nature* 473, 337-342.
- Smoak, E.M., Stein, P., Schultz, R.M., Lampson, M.A., and Black, B.E. (2016). Long-Term Retention of CENP-A Nucleosomes in Mammalian Oocytes Underpins Transgenerational Inheritance of Centromere Identity. *Curr Biol* 26, 1110-1116.
- Taher, L., Israel, S., Drexler, H.C.A., Makalowski, W., Suzuki, Y., Fuellen, G., and Boiani, M. (2021). The proteome, not the transcriptome, predicts that oocyte superovulation affects embryonic phenotypes in mice. *Sci Rep* 11, 23731.
- Zatsepina, O., Baly, C., Chebrout, M., and Debey, P. (2003). The step-wise assembly of a functional nucleolus in preimplantation mouse embryos involves the cajal (coiled) body. *Dev Biol* 253, 66-83.

Second round of review

Reviewer 1

My comments have been fully addressed.

I have enjoyed reading the manuscript and the point-by-point responses very much, and look forward to seeing it in the published format.

There is one issue introduced during the revision need to be corrected:

"It was reported that certain proteins are synthesized only once during the oocyte maturation..." shall be "It was reported that certain proteins are synthesized only once during the oocyte growth...".